# Empty conformers of HLA-B preferentially bind CD8 and regulate CD8+ T cell function

Jie Geng[1], John D Altman[2,3], Sujatha Krishnakumar[4], Malini Raghavan[1]*

[1]Department of Microbiology and Immunology, Michigan Medicine, University of Michigan, Ann Arbor, United States; [2]Department of Microbiology and Immunology, Emory University School of Medicine, Atlanta, United States; [3]Yerkes National Primate Research Center, Emory University, Atlanta, United States; [4]Sirona Genomics, Immucor, Inc., California, United States

**Abstract** When complexed with antigenic peptides, human leukocyte antigen (HLA) class I (HLA-I) molecules initiate CD8+ T cell responses via interaction with the T cell receptor (TCR) and co-receptor CD8. Peptides are generally critical for the stable cell surface expression of HLA-I molecules. However, for HLA-I alleles such as HLA-B*35:01, peptide-deficient (empty) heterodimers are thermostable and detectable on the cell surface. Additionally, peptide-deficient HLA-B*35:01 tetramers preferentially bind CD8 and to a majority of blood-derived CD8+ T cells via a CD8-dependent binding mode. Further functional studies reveal that peptide-deficient conformers of HLA-B*35:01 do not directly activate CD8+ T cells, but accumulate at the immunological synapse in antigen-induced responses, and enhance cognate peptide-induced cell adhesion and CD8+ T cell activation. Together, these findings indicate that HLA-I peptide occupancy influences CD8 binding affinity, and reveal a new set of regulators of CD8+ T cell activation, mediated by the binding of empty HLA-I to CD8.
DOI: https://doi.org/10.7554/eLife.36341.001

*For correspondence:
malinir@umich.edu

## Introduction

The major histocompatibility complex class I (MHC-I) molecules play a crucial role in adaptive immune responses by presenting antigenic peptides to CD8+ T cells, which enables the immune system to detect transformed or infected cells that display peptides from foreign or mutated self-proteins. Peptides are an integral component of MHC-I molecules. In the MHC-I antigen presentation process, peptides are mainly produced in the cytosol by proteasome and then translocated to the endoplasmic reticulum (ER) by the transporter associated with antigen processing (TAP). Peptides are loaded to MHC-I peptide binding groove with the assistance of several ER chaperones, ERp57, calreticulin and tapasin. There are several quality control components to ensure that most cell surface MHC-I molecules are filled with optimal peptide. However, under certain pathophysiological conditions, MHC-I peptide-deficient or open conformers are also detected on the cell surface. Prior evidence suggests that peptide-deficient conformers of MHC-I molecules appear on the cell surface of activated lymphoid cells (*Madrigal et al., 1991*; *Schnabl et al., 1990*), TAP-deficient cells (*Ljunggren et al., 1990*; *Ortiz-Navarrete and Hämmerling, 1991*) or EBV transformed B cells (*Madrigal et al., 1991*).

Although the presence of peptide-deficient conformers of MHC-I molecules on the cell surface under certain conditions is established, their functions are poorly understood. In the past few years, peptide-deficient conformers of MHC-I molecules of some allotypes have been shown to be ligands for cell surface receptors such as KIR3DS1 (*Burian et al., 2016*; *Garcia-Beltran et al., 2016*),

**eLife digest** The immune system keeps tabs on everything that happens in our body, looking for potential signs of threat. To alert it to any problems, almost every cell produces specific proteins on its surface called human leukocyte antigens class I, or HLA-I for short. These HLA-I molecules are bound to small protein fragments called peptides that have been exported from within the cell and are presented to the cells of the immune system for scanning. When cells are healthy, the peptides all stem from normal proteins. But, if the cell has become infected or cancerous, it contains foreign or abnormal peptides.

Some of the HLA-I molecules, however, are empty. These antigens are unstable, and their role is unclear. Now, Geng et al. investigated this further by studying blood samples from healthy donors. The experiments revealed that empty HLA-I molecules help specialized cells of the immune system, the killer T cells, to bind to the antigens, improving their killing ability.

It is known that these T cells recognize and bind to the antigens through two receptor proteins, one of which is called CD8. It was known that when HLA-I molecules carry a peptide, only a small fraction of T cells with a matching receptor can bind. However, Geng et al. found that when HLA-Is were empty, a much larger proportion of the T cells was able to bind to antigens. This indicates that CD8 'prefers' to attach to empty HLA-Is, maybe because binding sites are more accessible. CD8 also enhances the binding between the T cells and the antigen. Empty HLA-Is did not directly activate the T cells but did enhance their immune response. When both full and empty HLA-I were present, the T cells were even more effective at killing their targets.

Understanding how killer T cells work is essential for the development of immunotherapies – treatments that help to boost the immune system to fight infections and cancer. Increasing the number of empty HLA-I molecules on cancer or infected cells could enhance T cell killing.
DOI: https://doi.org/10.7554/eLife.36341.002

KIR3DL2 (*Goodridge et al., 2013*), KIR2DS4 (*Goodridge et al., 2013*) and LILRB2 (*Jones et al., 2011*). However, most of these studies involved a non-classical HLA-I, HLA-F (*Garcia-Beltran et al., 2016*), which has a higher propensity to be expressed in a peptide-deficient version compared to classical HLA-I molecules (*Goodridge et al., 2010*). The paucity of functional studies of peptide-deficient conformers of classic HLA-I could partly be attributed to their general low stability on the cell surface. In a previous study (*Rizvi et al., 2014*), we tested the refolding efficiencies of several HLA-B allotypes in the absence of peptide and found that peptide-deficient conformers of some allotypes such as B*35:01 are relatively more stable. Higher stability of peptide-deficient B*35:01 is also measurable in this study using a thermal unfolding assay with peptide-deficient HLA-B molecules that were engineered for enhanced stability via leucine zippered sequences (*Figure 1*). Peptide-receptive B*35:01 molecules are also detectable on the surface of activated T cells (this study) and TAP-deficient cells (Geng et al, submitted manuscript). Therefore, B*35:01 is a good representative HLA-B to investigate the function of peptide-deficient conformers of HLA-I molecules. In exploring potential binding partners for peptide-deficient conformers of HLA-I molecules, we found that tetramers of peptide-deficient conformers of HLA-B*35:01, in stark contrast to their peptide-filled conformer, stain a majority of blood-derived CD8[+] T cells. We hypothesized that the staining is largely CD8-mediated and also that peptide-deficient B*35:01 molecules can modulate CD8[+] T cell activation. Indeed, we show that CD8 prefers to bind peptide-deficient B*35:01 molecules and that peptide-deficient HLA-B*35:01 molecules on the cell surface enhance cell adhesion to CD8[+] T cells. Although they do not directly activate CD8[+] T cells, peptide-deficient HLA-B*35:01 molecules on the surface of antigen presenting cells enhance antigen-specific CD8[+] T cell responses. Together, these studies indicate key immune regulatory functions for peptide-deficient conformers of HLA-I molecules.

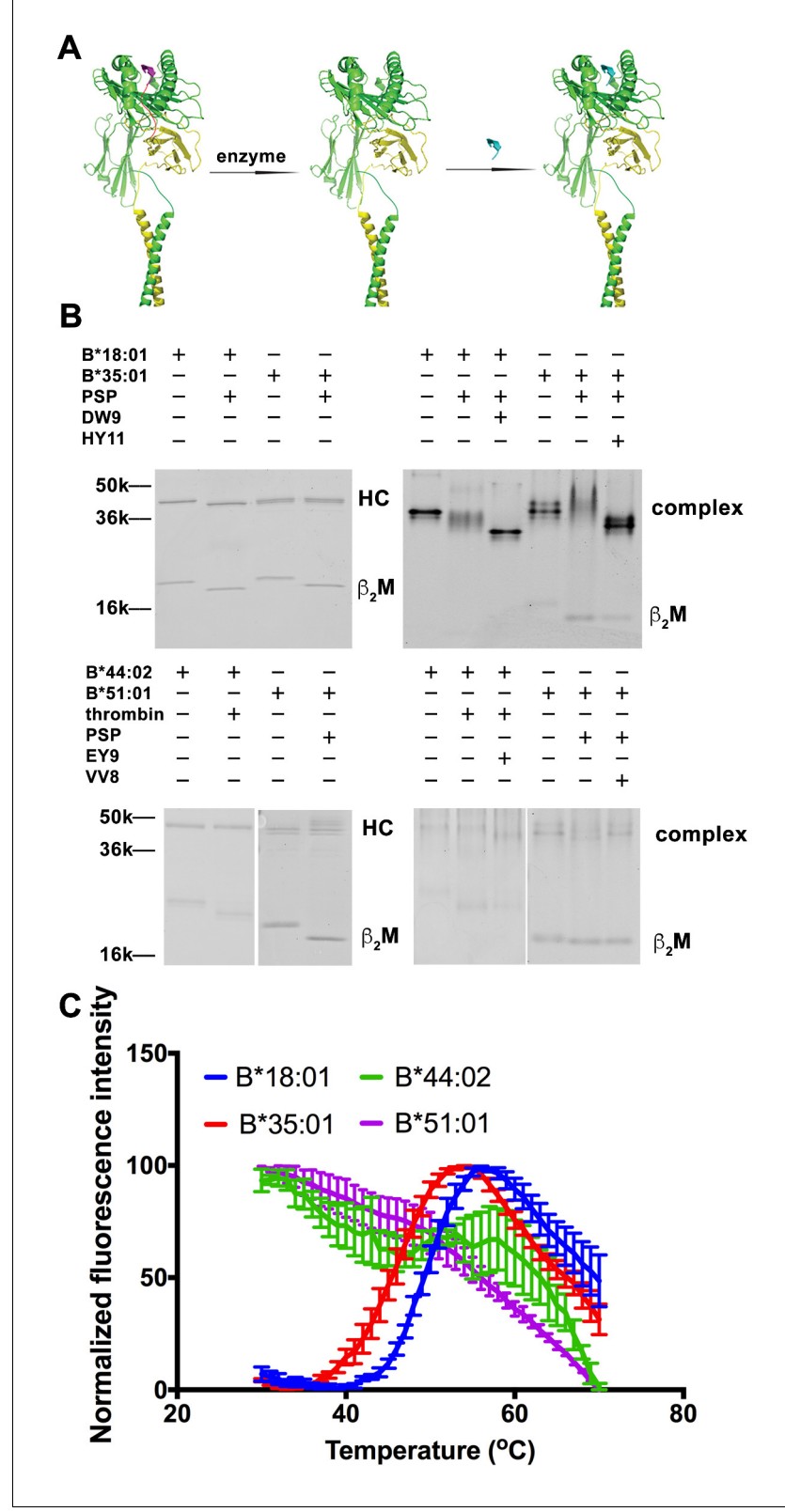

**Figure 1.** Peptide-deficient conformers of HLA-B molecules have different thermostabilities. (**A**) Peptide-deficient HLA-B can be prepared by cleavage of engineered HLA-B (LZ-ELBM) with a specific enzyme and peptide-loaded versions by subsequent incubation with specific peptide as described in the scheme. (**B**) Representative SDS (left panels) and native (right panels)-PAGE gels showed cleavage and loading of B*18:01, B*35:01, B*44:02, and

*Figure 1 continued on next page*

*Figure 1 continued*

B*51:01 molecules with peptides DEVASTHDW (DW9), HPVGEADYFEY (HY11), EEIPDFAFY (EY9) and VPYEPPEV (VV8), respectively. (C) Averaged (n $\geq$ 3 replicates) normalized thermal shift assays were performed with peptide-deficient conformers of B*18:01, B*35:01, B*44:02 and B*51:01 molecules. B*18:01 and B*35:01 are more stable than B*44:02 and B*51:01.

DOI: https://doi.org/10.7554/eLife.36341.003

## Results

### Higher relative thermostability of peptide-deficient conformers of some HLA-B allotypes

We previously quantified refolding efficiencies of HLA-B heterodimers based on in vitro refolding reactions conducted in the absence of peptides. Significant differences in folding efficiencies were noted (*Rizvi et al., 2014*). In the present study, we assessed whether peptide-deficient conformers of HLA-B allotypes also differ in their thermal unfolding characteristics (*Figure 1*). The NIH tetramer core facility has developed HLA-I molecules with epitope-linked β2m (ELBM), wherein an HLA-I binding peptide is covalently linked to human β2m via a linker peptide that contains a protease cleavage site. HLA-I heavy chain and β2m are further tethered via leucine zippers (LZ) at their C-termini (*Figure 1A*). Treatment of the HLA-I molecules with protease is expected to release the linked peptides, which are all C-terminally elongated, and thus sub-optimal for binding. When the cleavage is done in the presence of another HLA-I binding peptide, exchange should occur. Cleavage in the absence of peptide can produce peptide-deficient conformers of HLA-I molecules.

Peptide-deficient conformers of four HLA-B molecules, B*18:01, B*35:01, B*44:02 and B*51:01, were prepared and verified first by SDS-PAGE. The mobility of β2m was increased for all allotypes, consistent with expected reduction in molecular weight after cleavage (*Figure 1B* left panels). Formation and peptide loading of peptide-deficient conformers of HLA-B molecules were further validated by native-PAGE (*Figure 1B* right panels). In general, the mobilities for the cleaved HLA-B molecules are clearly different from uncleaved proteins. The reduced intensities of complex bands and increased intensities of the free β2m bands, likely because of heterodimer dissociation during electrophoresis, indicate that these heterodimers are less stable than uncleaved proteins. The HLA-B heterodimers were further loaded with allotype-specific peptides. The observed downward mobility shifts of the complex-specific bands for HLA-B*35:01, HLA-B*18:01 and HLA-B*44:02 in the native gels are indicative of peptide binding, and provide further evidence that the cleaved forms of those HLA-B molecules are in fact peptide-deficient. A downshift was not observed of the complex-specific bands for HLA-B*51:01, suggestive of low peptide-loading efficiency, also previously noted (*Rizvi et al., 2014*).

Thermostabilities of the peptide-deficient HLA-B molecules were assessed by comparing heat-induced unfolding with a thermal shift assay (*Figure 1C*). A fluorescent dye (Sypro Orange) was used that displays enhanced binding to proteins following thermal unfolding. Clear cut transitions are observable for peptide-deficient B*18:01 and B*35:01, but not for B*44:02 or B*51:01. These findings indicate important thermostability hierarchies among peptide-deficient conformers of HLA-B; allotypes such as B*35:01 and B*18:01 are more stable in their peptide-deficient conformers compared to allotypes such as B*44:02 and B*51:01, consistent with previously described refolding assay (*Rizvi et al., 2014*). As a representative peptide-deficient HLA-B with high thermostability, HLA-B*35:01 was used for further functional assessments.

### Peptide-deficient HLA-B*35:01 tetramers preferentially bind CD8 and stain CD8[+] T cells in a CD8-dependent manner

To explore potential receptors that are responsive to peptide-deficient HLA-B*35:01, tetramers were generated with peptide-deficient HLA-B*35:01 or their peptide-filled versions. Peripheral blood mononuclear cell (PBMC) staining of tetramers of peptide-deficient conformers was compared with the peptide-filled HLA-B*35:01. PBMCs obtained from healthy donors were stained with a panel of lymphocyte markers before tetramer staining. As expected, antigen-specific CD8[+] T cell populations were rare or absent in PBMCs from healthy B*35:01[+] donors, as assessed by staining with

uncleaved B*35:01 (carrying an epitope LPYPQPQPF from *Triticum aestivum*) tetramers (for example, *Figure 2A*). In contrast, peptide-deficient HLA-B*35:01 tetramers bound to most (over 70%) of total CD8+ T cells present in the donor (*Figure 2B*). These findings suggested that observed tetramer binding was unlikely to be linked to specific TCR. Rather, staining was significantly blocked by anti-CD8 (Clone SK1, BioLegend) (*Figure 2C*), suggesting that peptide-deficient B*35:01 tetramers are capable of binding to cell surface CD8 with higher potency compared to the peptide-filled version. In parallel analyses, CD4+ T cells were poorly stained by the peptide-deficient B*35:01 tetramers (*Figure 2D*) and staining was not blocked by anti-CD8 (*Figure 2E*), consistent with the finding of CD8-dependent binding to CD8+ T cells. Similar results were obtained with cells from a

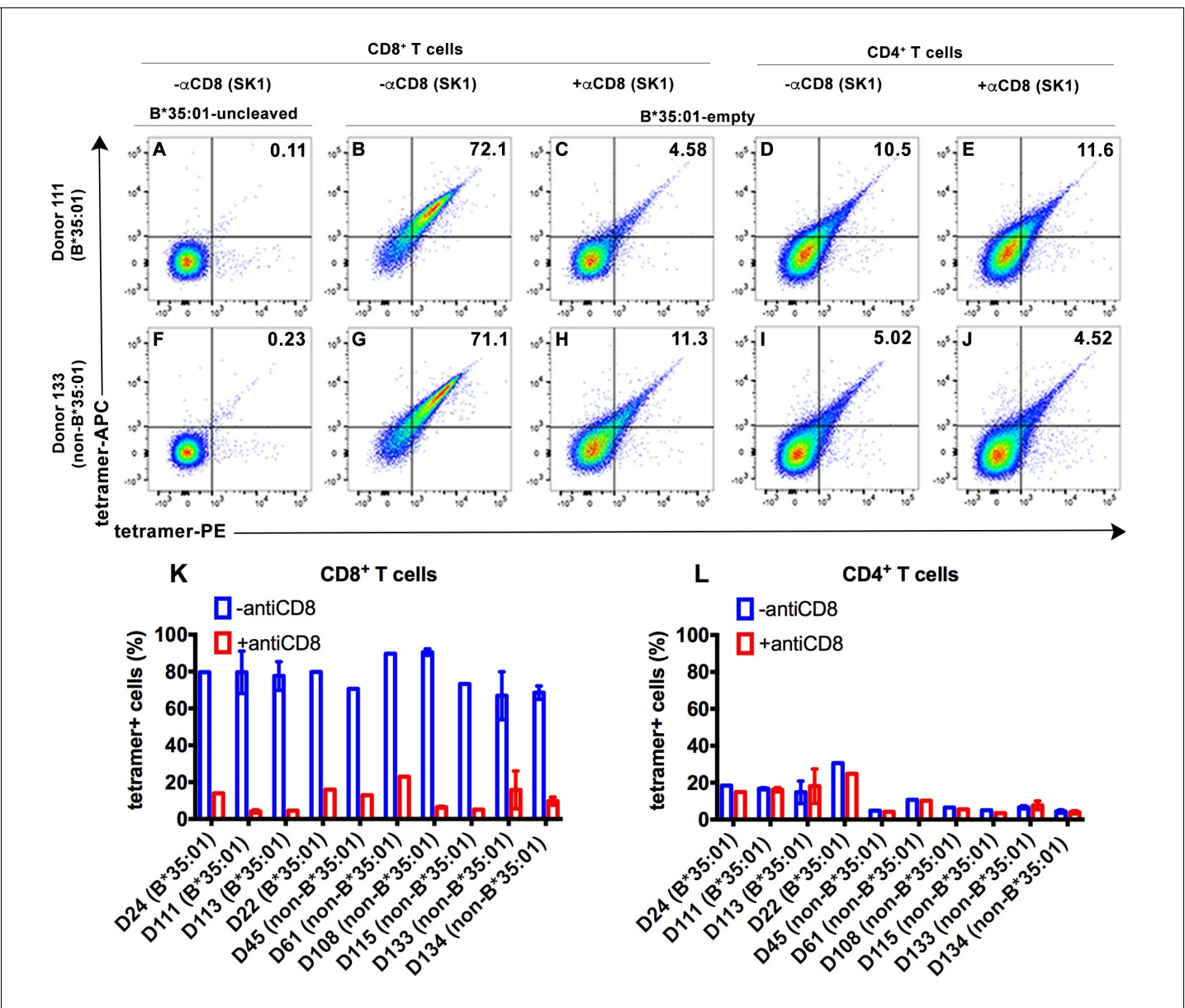

**Figure 2.** CD8-dependent binding of peptide-deficient HLA-B*35:01 tetramers to CD8+ T cells. Uncleaved B*35:01 tetramer poorly stained CD8+ T cells (A and F), whereas peptide-deficient B*35:01 tetramers stained most (over 70%) CD8+ T cells from B*35:01-positive donors (such as Donor 111) (B) or B*35:01-negative donors (such as Donor 133) (G) in a manner sensitive to blockage by anti-CD8 (Clone SK1) (C and H). CD4+ T cells were, in comparison, poorly stained by peptide-deficient B*35:01 tetramers and the staining cannot be blocked by anti-CD8 (D, E, I and J). Peptide-deficient B*35:01 tetramer staining data from 10 tested donors (mean ± SEM of one to two assays) are shown in (K and L).
DOI: https://doi.org/10.7554/eLife.36341.004

B*35:01-negative donor (*Figure 2F–J*), of enhanced CD8-dependent binding of peptide-deficient HLA-B*35:01 tetramers to CD8+ T cells (*Figure 2G–H*), and comparatively poor CD8-independent binding to CD4+ T cells (*Figure 2I–J*). Binding and inhibition data compiled from multiple B*35:01-positive and B*35:01-negative donors are shown in *Figure 2K and L*.

CD8 is also found on the surface of other lymphocytes, such as NK cells, although as a CD8αα homodimeric form instead of CD8αβ heterodimeric form (*Moebius et al., 1991*). We examined the ability of peptide-deficient B*35:01 tetramers to bind to CD8 on the NK cell surface. B*35:01 belongs to Bw6 serotype of HLA-B alleles that lack a binding sequence for engagement of the HLA-B recognizing killer immunoglobulin-like receptor, KIR3DL1 (*Gumperz et al., 1997*). In fact, we found that peptide-deficient B*35:01 tetramers exclusively bind to NK cells expressing CD8 and further that binding to the cells is fully blocked by anti-CD8 (*Figure 3A*). Furthermore, NK cells from different donors have different percentages of CD8 expressing NK cells, and the level of staining with peptide-deficient B*35:01 tetramers is directly proportional to the CD8 expressing fraction of NK cells (*Figure 3B*). Further experiments were undertaken to compare the binding of purified FITC-labeled CD8αα to peptide-deficient or peptide-filled HLA-B*35:01 conjugated to streptavidin resin (*Figure 3C*). Bead-bound CD8 was quantified by fluorimaging analyses of SDS-PAGE-separated samples. Nonlinear curve fitting analyses of the FITC fluorescence signals yielded a $K_D$ value of ~20 μM for peptide-deficient B*35:01, significantly stronger binding than that for peptide filled B*35:01, for which a $K_D$ value could not be accurately estimated (*Figure 3D*).

## Binding of CD8 to peptide-deficient HLA-B*35:01 enhances adhesion of CD8+ T cells to HLA-B*35:01 expressing TAP-deficient cells

CD8 can act as adhesion molecule, co-receptor and immuno-modulator (*Cole and Gao, 2004*). Interaction between MHC-I and CD8 is proposed to enhance cell adhesion (*Norment et al., 1988*). We assessed whether the stronger interaction between peptide-deficient HLA-B*35:01 and CD8 could enhance cell-cell adhesion. We expressed HLA-B*35:01 and a HLA-B*35:01 mutated at the CD8

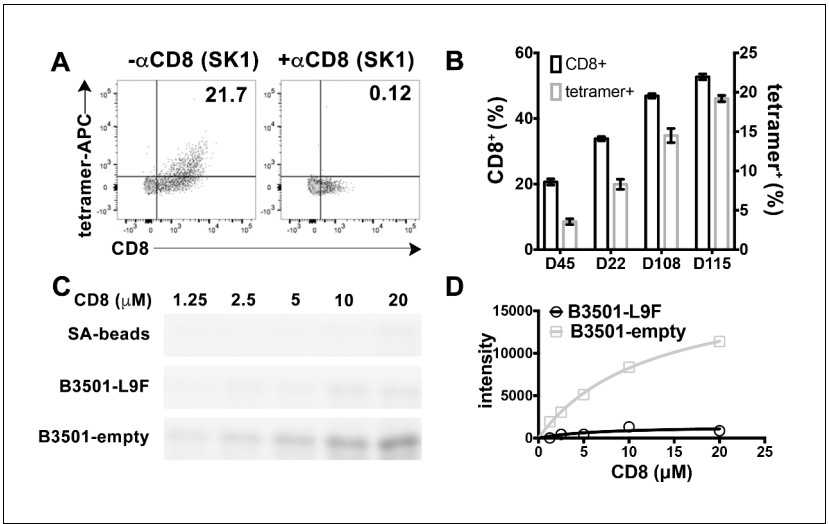

**Figure 3.** Preferential binding of peptide-deficient conformers of HLA-B*35:01 to CD8. (**A**) Primary NK cells (CD3-CD56+) from Donor 115 were stained with peptide-deficient B*35:01 tetramers, demonstrating specific binding to the CD8+ NK cell fraction (left panel). NK cell staining by peptide-deficient B*35:01 tetramers was blocked by anti-CD8 (right panel). Representative data are shown based on two experiments each with four donors. (**B**) Primary NK cells from different donors have different CD8+ fractions and CD8-dependent binding of peptide-deficient B*35:01 tetramer to NK cells is proportional to the CD8+ fraction of NK cells among tested donors. The mean ± SEM of two experiments for each donor are shown. (**C**) Binding of SA-bead immobilized peptide-deficient or peptide-filled B*35:01 to the indicated concentrations of CD8-FITC. Proteins pulled-down were analyzed by SDS-PAGE gel and fluorimaging. (**D**) Quantified binding signals are plotted following background subtraction. Data are representative of four experiments.

DOI: https://doi.org/10.7554/eLife.36341.005

binding residues (D227K/T228A; B*35:01-CD8 null) (*Purbhoo et al., 2001*), in a TAP1-deficient cell line SK19 (*Yang et al., 2003*). Both proteins are readily detectable on the cell surface (*Figure 4A–B*). Incubation with a B*35:01-specific peptide HPVGEADYFEY (HPV), but not a related truncated and mutated control peptide HGVGEADYFE (HGV), induces binding by the peptide-MHC-I complex-specific W6/32 antibody (*Parham et al., 1979*) and reduces binding by the heavy chain-specific HC10 antibody (*Stam et al., 1990*; *Gillet et al., 1990*) for both B*35:01 molecules (*Figure 4C–D*), indicating that at least a subset are able to be expressed as peptide-deficient conformers, under conditions where TAP, the major source of cellular MHC-I peptides, is absent.

To test cell adhesion mediated by peptide-deficient B*35:01 or its CD8-null version, two CTL lines were used as CD8 expressing cells. Cell conjugation between the SK19 cells and CTLs was investigated by two approaches, confocal microscopy and flow cytometry. In the microscopy assay, SK19 cells were pre-attached to glass-bottomed petri dish. After co-incubation, CTL line A2-AL9 specific for HLA-A*0201 complexed to the HIV-derived AL9 peptide (AIIRILQQL) (*Altfeld et al., 2001*) showed significantly increased adhesion to SK19 cells expressing HLA-B*35:01 compared with SK19 cells lacking HLA-B*35:01 (*Figure 4E–F*). There was also a very marked blocking effect of pre-incubation of SK19 HLA-B*35:01 cells with the B*35:01 specific peptide HPV upon conjugate formation (*Figure 4G*), suggesting that peptide-deficient B*35:01 is important for mediating cell adhesion. On the other hand, there was no significant cell adhesion enhancement with SK19-HLA-B*35:01-CD8 null compared with SK19 cells lacking HLA-B*35:01 (*Figure 4H*), reflecting the significance of CD8-B*35:01 binding upon enhancement of cell adhesion.

In the flow cytometry assays, SK19 cells were labeled with CFSE and then incubated with CTL. CFSE and CD8 double positive cell populations were identified as SK19-CTL conjugates, and used to quantify the effect of B*35:01-CD8 interactions on cell adhesion. Compared with SK19-HLA-B*35:01-CD8 null, SK19-HLA-B*35:01 showed stronger adhesion to CTL line, A2-AL9 (*Figure 4I* upper panel). Pulsing of SK19-HLA-B*35:01 with peptide HPV strongly reduced cell adhesion, consistent with the microscopy assays. We also generated a CTL line (B8-RL8) specific for HLA-B*08:01 complexed with the EBV-derived epitope RAKFKQLL (RL8), for use as second set of effector cells, and obtained similar results as with the A2-AL9 CTL line (*Figure 4I* lower panel).

## No direct activation of antigen-specific CD8+ T cell responses by peptide-deficient HLA-B*35:01

Stronger binding of peptide-deficient HLA-B to CD8 (*Figure 4*) raised the possibility of CD8+ T cell activation regulation by peptide-deficient conformers (*Wooldridge et al., 2010*). We therefore tested whether tetramers of peptide-deficient conformers directly activated CD8+ T cells. Intracellular staining assays were carried out to examine whether CD8 ligation with peptide-deficient B*35:01 tetramers induces cytokine expression. We found that cross-linking of CD8 with peptide-deficient B*35:01 tetramers at a concentration as high as 40 µg/ml failed to activate primary CD8+ T cells as well as the CTL lines A2-AL9 and B8-RL8, as assessed by the general absence of changes in the expression of cytokine interferon gamma (IFN-γ) (*Figure 4—figure supplement 1*). These findings suggested that the ligation of CD8 by peptide-deficient B*35:01 did not directly induce significant activation signaling.

## Peptide-deficient HLA-I is induced on activated CD4+ T cells and enriched in the antigen-dependent immunological synapse

Although ligation of CD8 by peptide-deficient B*35:01 did not induce direct activation of CD8+ T cells, the enhanced cell-adhesion mediated by the interaction (Figure 2-*Figure 4*) could modulate antigen-specific CD8+ T cell activation. This could not be tested in TAP-deficient SK19 cells, due to very low cell surface expression of both HLA-A*02:01 and HLA-B*08:01, the HLA-I molecules recognized by the two CTL lines used in *Figure 4*. Lymphocyte activation is previously shown to induce forms of HLA-I molecules that are recognized by HC10 (*Matko et al., 1994*), the antibody specific for peptide-deficient conformations (*Stam et al., 1990*). Consistent with these prior findings, purified CD4+ T cells from different donors were shown to consistently induce HC10-reactive HLA-I molecules on the cell surface following their activation with PHA (*Figure 5A*). As discussed below in *Figure 6*, the presence of peptide-receptive B*35:01 was directly measurable on activated CD4+ T cells from HLA-B*35:01+ donors. Activated CD4+ T cells were thus used as antigen-presenting cells

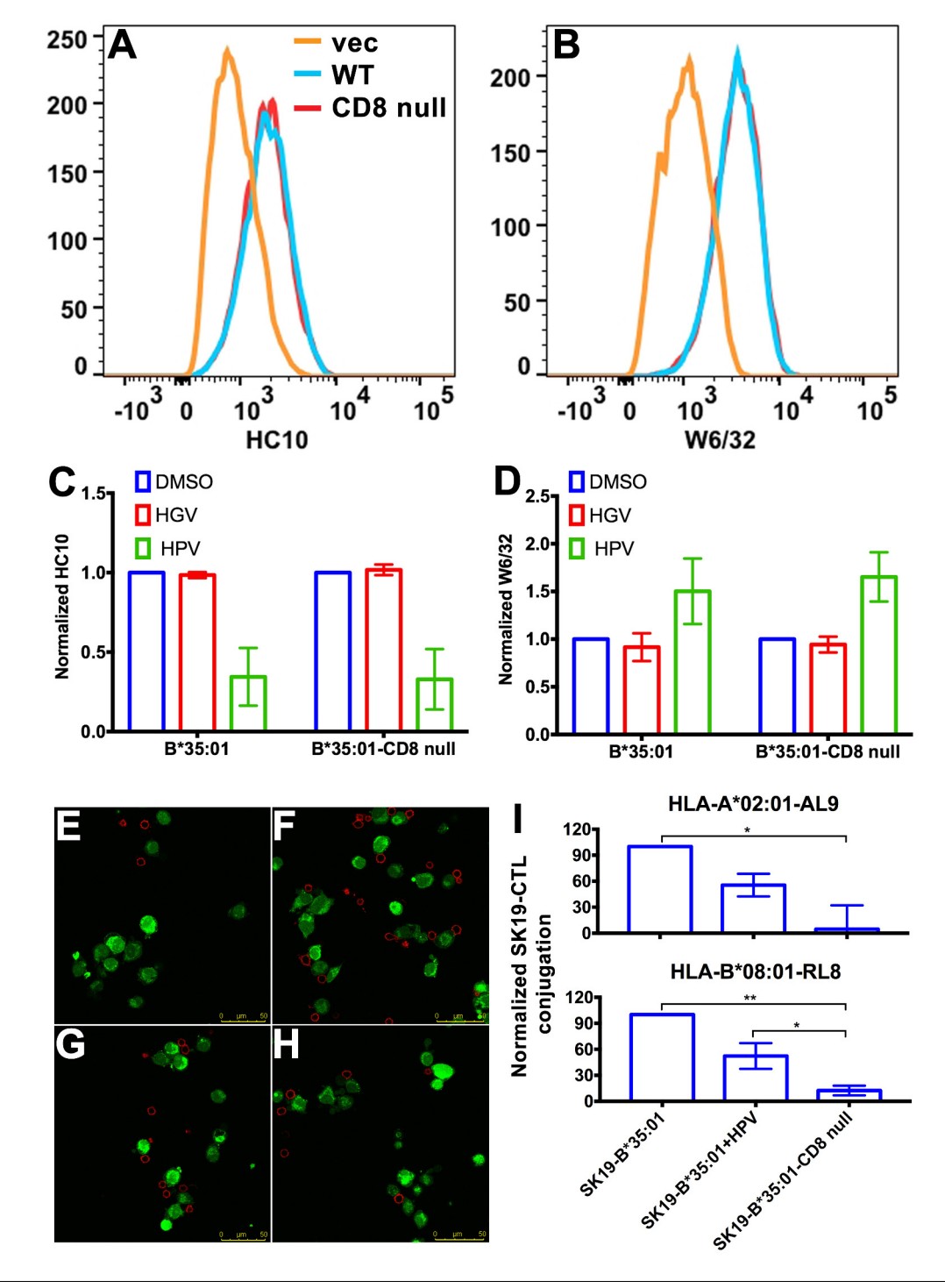

**Figure 4.** Binding of peptide-deficient conformers of HLA-B*35:01 to CD8 enhances cell adhesion. HLA-B*35:01 and HLA-B*35:01-CD8 null were expressed by retroviral infection in the TAP1-deficient cell line, SK19. Similar levels of HLA-I in either peptide-deficient (**A**) or peptide-filled (**B**) versions were detected on the cell surface by flow cytometry. The peptide-deficient conformers can partly be blocked by the HLA-B*35:01-specific peptide (HPV) but not control peptide (HGV), which are indicated by reduced HC10 staining (**C**) and enhanced W6/32 staining (**D**). The mean ± SEM of two experiments are shown. Confocal microscopy (**E–H**) was used to test cell adhesion between SK19 cells expressing HLA-B*35:01 or HLA-B*35:01-CD8 null and a CTL line A2-AL9. A2-AL9 was incubated with preattached and CFSE-labeled SK19 cells (green) infected with retroviruses lacking HLA-B (**E**),

*Figure 4 continued on next page*

*Figure 4 continued*
or encoding HLA-B*35:01 (F and G) or HLA-B*35:01-CD8 null (H). For G, SK19-HLA-B*35:01 cells were preloaded with peptide HPV (100 µM). Cells were washed, fixed and stained with anti-CD8 (red) before analysis. Representative data are shown. Flow cytometry was used as a more quantitative assessment to test cell adhesion between SK19 cells expressing HLA-B*35:01 or HLA-B*35:01-CD8 null and CTL lines, A2-AL9 or B8-RL8 (I). CFSE and CD8 double positive cells were quantified as percentages of total SK19 cells. The condition with SK19 cells lacking HLA-B was subtracted as background. The mean ± SEM of three experiments are shown. Statistical analyses were undertaken using one-way ANOVA analysis with Fisher's LSD test. *p<0.05, **p<0.01.
DOI: https://doi.org/10.7554/eLife.36341.006
The following figure supplement is available for figure 4:

**Figure supplement 1.** No direct activation of peptide-deficient HLA-B*35:01 tetramers on CTL activation.
DOI: https://doi.org/10.7554/eLife.36341.007

to present peptide-HLA–B*08:01 or peptide-HLA-A*02:01 complexes, and provide a parallel source of peptide-deficient B*35:01 for further functional assessments of the effects of peptide-deficient B*35:01 on antigen-specific T cell responses.

Molecular clustering within the immunological synapse is emerging as key mechanism for the control of T cell activation. Therefore, we first used a cell-cell contact assay to determine whether peptide-deficient conformers are clustered within the immunological synapse induced by recognition of RL8-HLA–B*08:01 by the B8-RL8 CTL line. CD4$^+$ T cells from a donor expressing both HLA-B*08:01 and HLA-B*35:01 were pre-activated to induce peptide-deficient conformers on the cell surface. B8-RL8 CTLs were co-incubated with the activated CD4$^+$ T cells pulsed with the antigenic peptide RL8. RL8 induces stronger clustering of HLA-I peptide-deficient conformers (measured with the peptide-deficient conformer-specific antibody HC10, *Figure 5D–E,H*) than peptide-filled HLA-I (measured with the peptide-MHC-I complex-specific W6/32 antibody, *Figure 5B–C and H*) in the interface between antigen presenting cells (APC) and CTLs. On the other hand, in the absence of RL8 peptide, cell conjugates were strongly reduced and little enrichment was observed in the junctions between CTL and APC (*Figure 5F–G,H*). We did not observe strong CD8 clustering in the interface, probably due to different kinetics of MHC-I and CD8 clustering which has been reported previously (*Purbhoo et al., 2004*). Similar findings were obtained when activated PBMC rather than activated CD4$^+$ T cells were used as the antigen-presenting cells (*Figure 5—figure supplement 1*).

## HLA-B*35:01 peptide-deficient conformers enhance cognate peptide-induced antigen-specific target cell lysis

We further tested whether the peptide-deficient HLA-B*35:01-CD8 interaction has a regulatory effect on cognate antigen-induced target cell lysis. The CTL line specific to the HLA-A2-AL9 complex was first chosen as the effector cell. Primary CD4$^+$ T cells expressing both A*02:01 and B*35:01 were used as target cells. After incubation at effector-to-target ratios of 1:1, 5:1 and 20:1, CTLs exhibited a strong increase in the ability to kill target cells pulsed with the cognate peptide AL9 (*Figure 6—figure supplement 1*, right panels) compared to target cells pulsed with control peptide SLYNTVATL (SL9) (*Figure 6—figure supplement 1*, left panels). Next, primary CD4$^+$ T cells expressing both A*02:01 and B*35:01 from Donor 24 were activated and cell surface expression of peptide-deficient conformers were detected with peptide-deficient conformer-specific antibody HC10. Peptide-deficient conformers could be partially blocked by B*35:01-specific peptides YPLHEQHGM (YPL), HPNIEEVAL (HPN), HPV and FPTKDVAL (FPT), but not control peptide (TSTLQEQIGW, TW10) (*Figure 6F–I*), indicating that a subset of the HLA-B*35:01 molecules are peptide-deficient. We found that the cognate peptide-induced CD4$^+$ T cell lysis can partly be blocked by B*35:01-specific peptides (*Figure 6A*), suggesting that HLA-B*35:01 peptide-deficient conformers do enhance cell lysis induced by cognate peptides. The effect on modulating antigen-specific cell lysis is observed with different HLA-B*35:01 peptides and across different donors (*Figure 6B–C*). Similar effects were also observed in cell lysis assays with the B8-RL8 CTL line (*Figure 6D–E*).

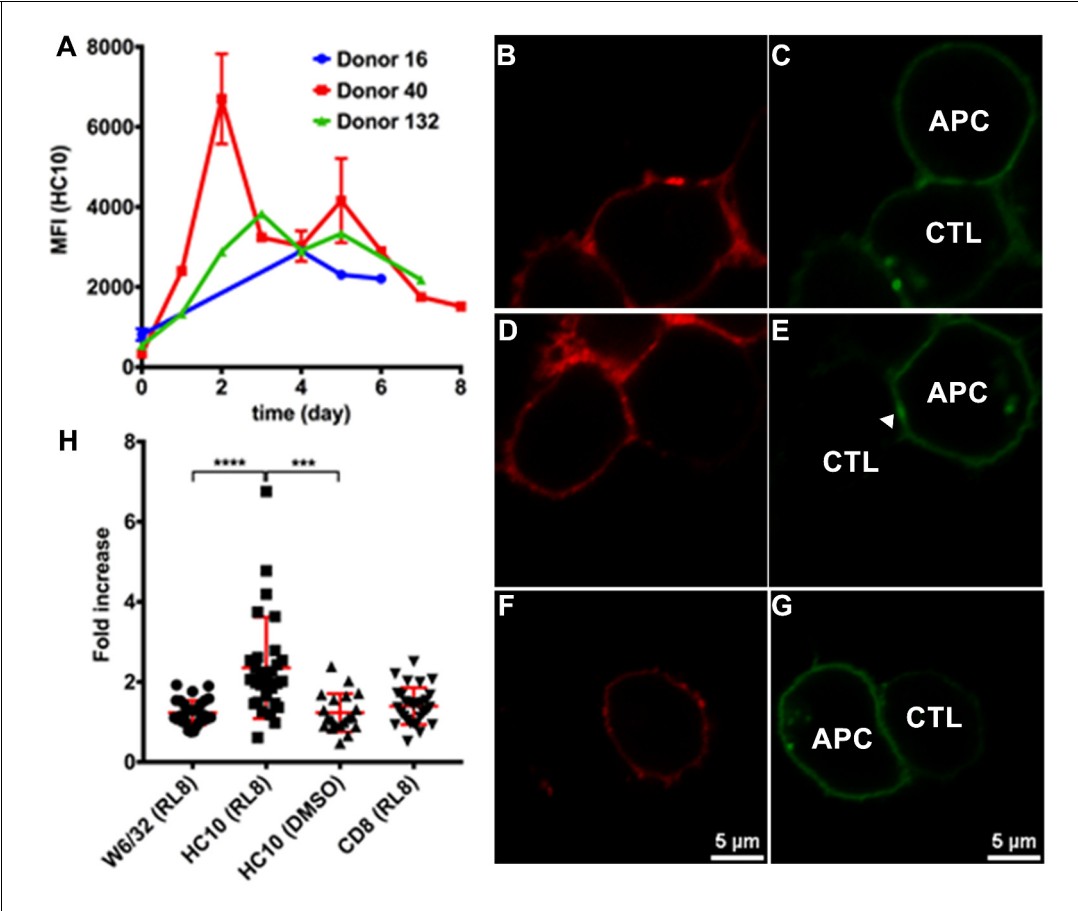

**Figure 5.** Clustering of peptide-deficient HLA-I in cognate peptide-induced immunological synapses. Expression levels of HLA-I on the surface of activated CD4+ T cells isolated from PBMCs of three healthy donors were assessed by flow cytometry after staining with HC10 (**A**), confirming that activated lymphocytes express peptide-deficient conformers of HLA-I molecules. The mean ± SEM of two replicates for each donor are shown. CTL line B8-RL8 was incubated with activated CD4+ T cells from Donor 25 (carrying B*08:01 and B*35:01) loaded with peptide RL8 (**B**, **C**, **D** and **E**) or not (**F** and **G**). Cells were fixed and stained with anti-CD8 and W6/32 or HC10 before analysis by confocal microscopy. Peptides were used at a concentration of 100 μM. Anti-CD8 staining (**B**, **D** and **F**) were shown in red and W6/32 (**C**) or HC10 (**E** and **G**) in green. Arrowheads indicate peptide-deficient conformers of HLA-I clustering at the interface between CD4+ T cells and CTL line. The intensity of HLA-I staining of the CD4+ T cells at the interface was compared with the membrane at a noncontact area and plotted as the fold increase above background (**H**). The results with a total of 20–30 conjugates (mean ± SEM) per condition are shown. CD8 clustering was derived from 30 conjugates under conditions shown in D. Statistical analyses were undertaken using a one-way ANOVA analysis with Fisher's LSD test. ***, p<0.001; ****, p<0.0001.
DOI: https://doi.org/10.7554/eLife.36341.008

The following figure supplement is available for figure 5:

**Figure supplement 1.** Clustering of peptide-deficient HLA-I in cognate peptide-induced immunological synapses.
DOI: https://doi.org/10.7554/eLife.36341.009

## Discussion

The major functions of MHC-I proteins include presenting antigenic peptides to CD8+ T cells and delivering activation or inhibitory signals to NK cells. It was widely known that the interactions between MHC-I molecules with their receptors are both allotype and peptide dependent. Our studies indicate that peptide-deficient MHC-I molecules are also functional in the immune response. A given peptide-HLA-I complex is typically able to engage only a small percentage of blood-derived CD8+ T cells, those that bear an appropriate TCR. In contrast, peptide-deficient conformers of HLA-B*35:01 engaged a majority of CD8+ T cells from multiple donors (*Figure 2*). While the pMHC-I/CD8 interaction is generally characterized by very low affinities (*Wang et al., 2009*; *Wyer et al., 1999*), we find that peptide-free HLA-B*35:01 binds CD8 with significantly higher affinity than their peptide-filled versions (*Figure 3*). Thus, unlike peptide-occupied HLA-B that engage CD8+ T cells via a

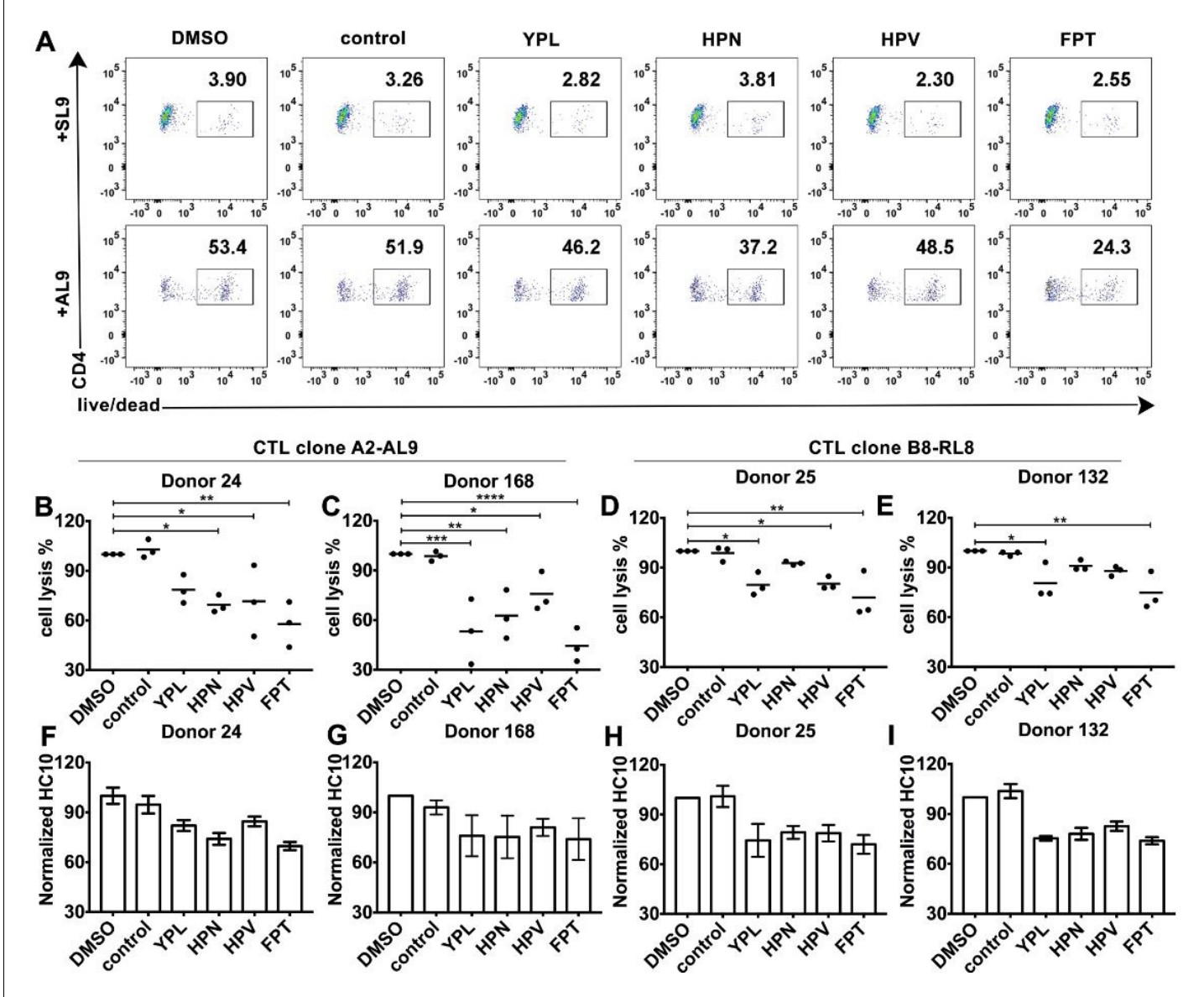

**Figure 6.** HLA-B*35:01 peptide-deficient conformers enhance cognate peptide-induced lysis of target cells by CD8[+]T cell activation. CTL line A2-AL9 specifically killed activated primary CD4[+] T cells (expressing HLA-A*02:01 and HLA-B*35:01 and used as antigen presenting cells) loaded with cognate peptide AL9 but not non-cognate peptide SL9 (A). Peptides that binds specifically to HLA-B*35:01 (YPL, HPN, HPV and FPT) could reduce AL9-induced cell lysis. Representative data with CD4[+] T cells from Donor 24 are shown in Panel A and statistical analyses from Donors 24 and 168 in Panels B and C. Cell lysis assays were also performed with another CTL line B8-RL8 and activated primary CD4[+] T cells (carrying HLA-B*08:01 and B*35:01) from Donors 25 and 132 loaded with cognate peptide RL8 were used as target cells (D and E). (B–E) The mean ± SEM of three independent assays are shown. Statistical analyses were undertaken using one-way ANOVA analysis with Dunnett test. *, $p<0.05$, **, $p<0.01$, ***, $p<0.001$, ****, $p<0.0001$. (F–I) Blocking of HLA-B*35:01 peptide-deficient conformers on the surface of CD4[+] T cells by B*35:01-specific peptides (YPL, HPN, HPV and FPT), but not nonspecific control peptide (TW10 or HGV) was confirmed by flow cytometry after staining cells with HC10. The mean ± SEM of two to three independent experiments are shown.

DOI: https://doi.org/10.7554/eLife.36341.010

The following figure supplement is available for figure 6:

**Figure supplement 1.** Cell lysis by CTL line A2-AL9 of activated primary CD4[+] T cells expressing HLA-A*02:01.

DOI: https://doi.org/10.7554/eLife.36341.011

TCR–dependent binding mode, peptide-deficient conformers of HLA-B*35:01 engage CD8⁺ T cells via a binding mode that is largely CD8-dependent.

The MHC-I-binding site for CD8 is spatially separated from the peptide-binding domains that are recognized by the TCR, and this spatial segregation allows both TCR and CD8 to bind a single MHC-I molecule simultaneously. In contrast to peptide-loaded MHC-I molecules, peptide-free MHC I molecules are suggested to possess properties similar to molten globules (*Bouvier and Wiley, 1998*), and show more protein plasticity based on MD simulations (*van Hateren et al., 2013*). The stronger binding to CD8 of peptide-deficient HLA-B compared to peptide-filled HLA-B (*Figure 3*) is likely caused by conformational differences between peptide-occupied and peptide-deficient conformers of HLA-B molecules that determine the accessibility or orientation of the CD8 binding site on HLA-I. Peptide-deficient HLA-I molecules are also preferred by ER chaperones tapasin and TAPBPR, which functions to facilitate peptide loading of MHC-I molecules. Crystal structures of tapasin-MHC-I and TAPBPR-MHC-I complexes highlight some common MHC-I binding sites by tapasin/TAPBPR and CD8 (*Blees et al., 2017*; *Jiang et al., 2017*; *Thomas and Tampé, 2017*). Residues at the C-terminal immunoglobulin-like domain of tapasin are positioned close to the CD8 recognition loop (especially residues 225 and 226) of the α3-domain of the MHC-I heavy chains (*Gao et al., 1997*; *Wang et al., 2009*), suggesting that the sites co-evolved (*Blees et al., 2017*). A β hairpin of TAPBPR at the N-terminal domain, which reaches under the floor of the peptide-binding groove, is important for sensing the conformation changes of the peptide-binding groove (*Thomas and Tampé, 2017*). CD8 also interacts with MHC-I at a similar region (including residues 115, 122 and 128) (*Gao et al., 1997*). Peptide loading reduces binding affinity between MHC-I molecules from tapasin and TAPBPR resulting in release of MHC-I molecules from tapasin and TAPBPR (*Rizvi and Raghavan, 2006*; *Wearsch and Cresswell, 2007*). CD8 might share a similar mechanism as tapasin and TAPBPR to distinguish MHC-I molecules with different conformations.

CD8 functions as an adhesion molecule and co-receptor to enhance the formation of TCR/pMHC complexes and the activation of CD8⁺ T cells. Although ligation of CD8 with non-cognate peptide-MHC-I complex was proposed to augment CD8⁺ T cell activation levels (*Anikeeva et al., 2006*; *Yachi et al., 2005*), generally, cognate peptide loading of MHC-I molecules is indispensable for CD8 T cell activation. Although a previous study showed that MHC-I molecules with super-enhanced CD8 binding properties bypass the requirement for cognate TCR recognition and nonspecifically activate CTLs (*Wooldridge et al., 2010*), we did not see any direct activation of CTL by HLA-B*35:01 peptide-deficient conformers. Nonetheless, our data suggested that preferential engagement of CD8 by peptide-deficient conformers of HLA-B*35:01 enhances cognate peptide-induced cell lysis (*Figure 6*). The enhancement of T cell activation appears to be caused by enhanced cell adhesion induced by the peptide-deficient conformer-CD8 interaction, or enhanced signaling induced by peptide-deficient conformer enriched within the immunological synapse (*Figure 7*). To escape immune surveillance by CD8⁺ T cells, several pathogens and tumors block HLA-I antigen presentation pathways to prevent antigenic peptide presentation by HLA-I molecules. Interestingly, many pathogen evasion or tumor progression strategies involve the targeting of the TAP transporter, inducing the cell surface expression of partially peptide-deficient HLA-I, as illustrated for HLA-B*35:01 (*Figure 4*). Peptide-deficient conformers of HLA-I are expected to enhance CD8⁺ T cells responses against TAP-independent epitopes, and thus counter the pathogen evasion strategies that target the HLA-I antigen presentation pathway.

Further studies are needed to quantitatively understand the extent of allele-dependent variations in CD8 binding by both peptide-deficient and peptide-filled conformers of HLA-B, as well as the induction of HLA-I peptide-deficient conformers under different physiological and pathological conditions. Peptide-deficient forms of different HLA-B allotypes were shown to have distinct thermostabilities and are therefore expected to be expressed at different levels on the cell surface. HLA alleles are known to differently associate with disease progression outcomes in major infectious diseases such as acquired immune deficiency syndrome (AIDS) (*Carrington and Walker, 2012*) and with autoimmune diseases such as ankylosing spondylitis (AS) (*Brown et al., 2016*), but the general underlying mechanisms are incompletely characterized. AS has been linked to the expression of HLA-B*27:05-free heavy chains (*Khare et al., 1996*), which can readily be detected on the surface of TAP-deficient cells (*Allen et al., 1999*). It would be of interest to test whether the interactions between peptide-deficient conformers of HLA-B*27:05 molecules and CD8 are involved in the onset and outcome of these diseases.

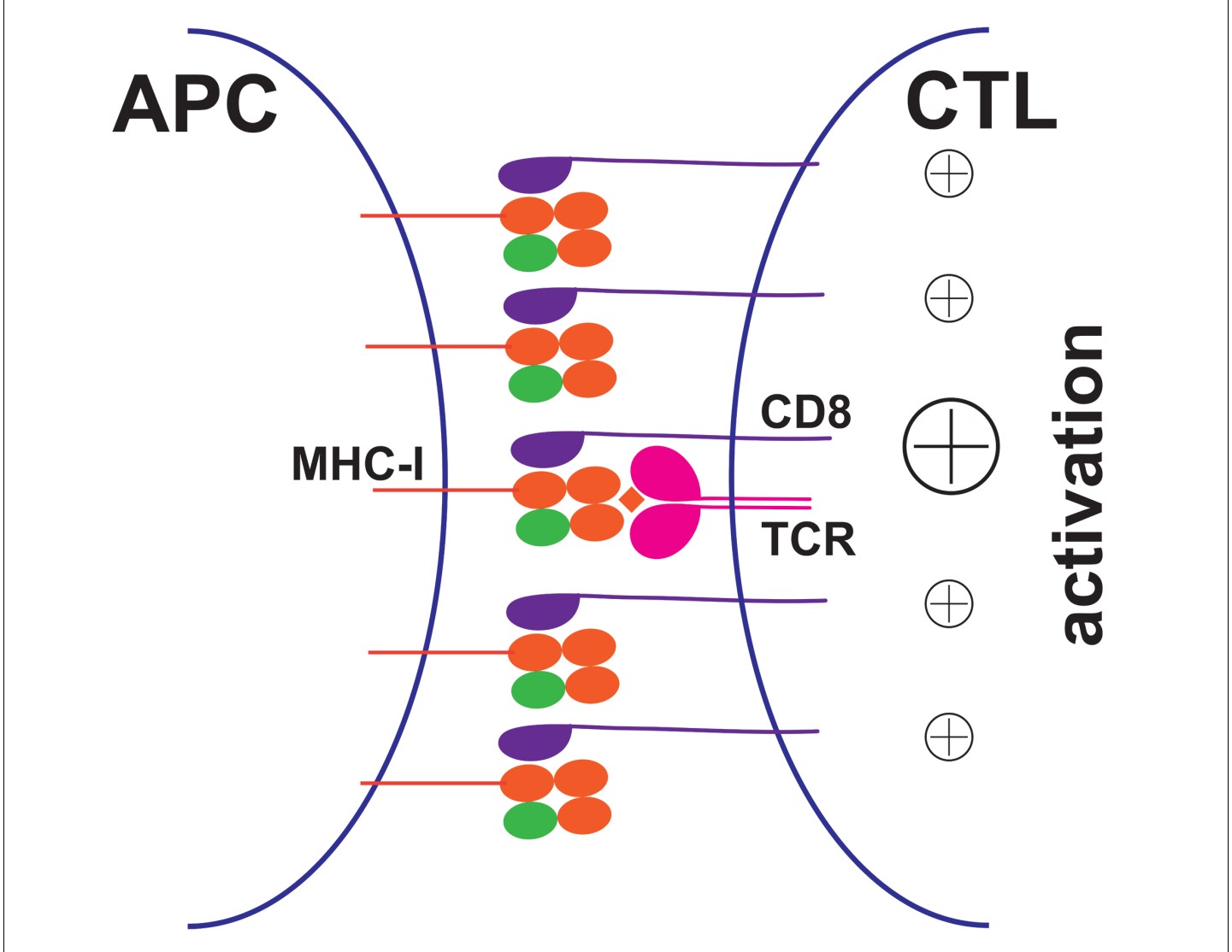

**Figure 7.** Peptide-deficient conformers of MHC-I molecules enhance CTL activation. CTL activation is generally induced by the recognition of a specific peptide-MHC-I complex on the surface of antigen presenting cells (APC) by a T cell receptor (TCR) and co-receptor CD8. In this study, we show that peptide-deficient conformers of MHC-I molecules have increased binding to CD8 compared with peptide-filled conformers and therefore enhance the cell-cell contact between APC and CTL. Peptide-deficient conformers of MHC-I molecules are enriched in the immunological synapse and augment CTL activation.

DOI: https://doi.org/10.7554/eLife.36341.012

In conclusion, our findings indicate that, without interaction with TCR, the peptide-deficient conformers of HLA-B*35:01 are able to interact efficiently with CD8. The preferential interaction between HLA-I peptide-deficient conformers and CD8 described here identifies a previously unknown mechanism by which CTL can be regulated. Finally, HLA-B peptide-deficient conformer-CD8 interactions may also have physiological regulatory functions in NK cell biology, which requires further study.

## Materials and methods

### Study approval

Blood was collected from consented healthy donors for HLA genotyping and functional studies in accordance with a University of Michigan IRB approved protocol (HUM00071750).

### Cell lines

Human melanoma cell line SK-mel-19 (SK19) (RRID: CVCL_6025) (*Yang et al., 2003*) and ecotropic virus packaging cell line BOSC (RRID: CVCL_4401) were grown in DMEM (Life Technologies) supplemented with 10% (v/v) FBS (Life Technologies) and 1 × Anti/Anti (Life Technologies) (D10). SK19 cells were gifted by Dr. Pan Zheng and verified for the absence of TAP1 expression. BOSC cells were obtained from the lab of Dr. Kathleen Collins. CTL line A2-AL9 was kindly gifted by Dr. Bruce Walker. CTL line B8-RL8 was generated in the lab by sorting after tetramer staining as previously described (*Dong et al., 2010*).

### Antibodies

The following monoclonal antibodies were used in this study: Ascites of W6/32 and HC10 from the University of Michigan Hybridoma Core, purified anti-human CD8a (clone SK1; BioLegend), AF700-conjugated anti-human CD8a (clone HIT8a; BioLegend), APC-Cy7-conjugated anti-human CD4 (clone RPA-T4; BioLegend), Pacific Blue-conjugated anti-human CD3 (clone UCHT1; BioLegend), PE-Cy7-conjugated anti-human CD56 (clone CMSSB; eBioscience), purified anti-human CD28 (clone 28.2; BD Biosciences) and FITC-conjugated anti-human IFN-γ (clone 4S.B3; BD Biosciences). Dead cells were excluded from flow cytometric analyses with 7-amino-actinomycin D (7-AAD; BD Biosciences) or the amine-reactive dye aqua (405 nm, Life Technologies).

### Isolation of peripheral blood mononuclear cells (PBMC)

Fresh blood was subjected to centrifugation over a Ficoll-Paque Plus (GE Healthcare Life Sciences) density gradient, washed twice in PBS and resuspended in RPMI1640 (Life Technologies) supplemented with 10% (v/v) FBS (Life Technologies) and 1 × Anti/Anti (Life Technologies) (R10). Assays were performed either on freshly isolated PBMC used within 2 to 4 hr of cell preparation, or on PBMC cryopreserved in Recovery Cell Culture Freezing Medium (Life Technologies).

### HLA typing

DNA was extracted from PBMCs using DNeasy Blood and Tissue Kit (Qiagen). The HLA typing was performed by Sirona Genomics (Mountain View, CA), an Immucor Company. The assay, based on a previous publication (*Wang et al., 2012*) was performed using the MIA FORA NGS HLA typing assay for the class I loci. The full-length amplicons for the class I loci were amplified and pooled. These samples were then fragmented, and tagged with unique index adaptors. The samples were pooled and sequenced on the Illumina MiSeq, and the HLA type was determined using the MIA FORA NGS HLA typing software. The Sirona Genomic HLA typing method has been validated by the Histocompatibility, Immunogenetics and Disease Profiling Laboratory of the Stanford University School of Medicine using 50 reference cell lines.

### Thermal shift assay

LZ-ELBM HLA-B molecules were provided by the NIH tetramer core facility. Peptide-deficient conformers of molecules were prepared by incubation with PreScission protease or thrombin overnight at 25°C. Cleaved fragments were removed by centrifuging the sample in a 0.5 ml Amicon Ultra filter device for 30 min at 13,000 rpm, 4°C. Native-PAGE and SDS-PAGE gels were both run to verify that the cleavage was efficient and HLA-B molecules became peptide-deficient. Peptide exchanges were performed by incubating HLA-B molecules with high affinity peptides together with PreScission protease or thrombin overnight at 25°C. Thermal shift assays were undertaken as previously described (*Del Cid et al., 2010*; *Huynh and Partch, 2015*). HLA-B molecules (8 μM) were incubated in buffer (PBS, pH7.4) and 1 × Sypro Orange Stain (Invitrogen) in a total reaction volume of 20 μl. Thermal scans were performed using an ABI PRISM 7900HT Sequence Detection System with temperature

increments of 1°C. Fluorescence emission was measured at ROX channel. Fluorescence was normalized within wells as percent maximum fluorescence and plotted against the sample temperature.

## Tetramer staining

Peptide-deficient conformers of HLA-B*35:01 were prepared by treatment of LZ-ELBM HLA-B*35:01 molecules with PreScission protease (GE Healthcare Life Sciences) for 2 hr at room temperature to release the tethered peptide, while peptide exchange was performed by adding HLA-B*35:01 binding epitopes simultaneously with PreScission protease. HLA-B*35:01 molecules were further dialyzed thoroughly with Amicon Ultra Centrifugal Filter Devices (Millipore) with a 10 kDa cutoff to remove unbound peptides. The peptide-deficient and peptide-exchanged monomers were verified by SDS or native-PAGE gels. The peptide epitope HPV was used in this study to reconstitute B*35:01. Tetrameric HLA-I reagents were constructed by the addition of streptavidin conjugated to PE (Prozyme, PJRS25) or APC (Prozyme, PJ27S) at 4:1 molar ratios following the tetramerization protocol from NIH core facility. PBS + 0.5% dialyzed BSA was used as staining buffer. For CD8 blocking, anti-CD8 (clone SK1; BioLegend) was incubated with PBMCs at 10 µg/ml for 15 min at room temperature. After washing once, freshly prepared tetramers were added typically at 20 µg/ml and anti-CD8-AF700 (clone HIT8a; BioLegend), anti-CD3-pacific blue (clone UCHT1; BioLegend) and anti-CD4-APC-Cy7 (clone RPA-T4; BioLegend) were added at concentrations indicated by the manufactures and incubated for another 30 min at room temperature. Cells were washed 3 times and 7AAD was added as a live/dead marker and samples were then analyzed by flow cytometry on a BD FACS-Canto II flow cytometer. The FACS data were analyzed with FlowJo software version 10.0.8 (Tree Star, San Carlos, CA).

## In vitro HLA and CD8 binding assay

Soluble human CD8αα (residues 1–120) with his-tag at the N-terminus was expressed in *Escherichia coli*, refolded, and purified by gel filtration as a ~30 kDa homodimer (*Gao et al., 1997*). The CD8αα concentration was calculated from the extinction coefficient, which was determined by amino acid analysis to be 37150 $M^{-1}cm^{-1}$ at 280 nm. CD8αα was labeled with FITC according to manufacturer's protocol (Thermo Scientific, **Rockford, IL, USA**). Relevant biotinylated HLA-B monomers from the tetramer core were immobilized onto streptavidin-coated agarose resin. FITC-labeled soluble CD8αα was added at different concentrations (2.5, 5, 10 and 20 µM) to immobilized HLA-B in binding buffer (PBS + 0.5% BSA). CD8αα was pulled down after co-incubation with the resin, and the beads were washed with binding buffer. SDS loading buffer was added and samples were denatured by heating for 10 min. Samples were loaded and resolved by SDS-PAGE and visualized by fluorimaging on a Typhoon scanner (at 520 nm). The binding at each concentration was obtained by subtraction of the control response (resin alone) from the B*35:01 response.

## Viruses and cell infections

HLA-B*35:01 in retroviral vector LIC pMSCVneo were prepared as described previously (*Rizvi et al., 2014*). The HLA-B*35:01 mutant that cannot bind CD8 (HLA-B*35:01-CD8 null) was made by introducing D227K/T228A mutations into HLA-B*35:01 using the QuikChange II site-directed mutagenesis kit. The primers were 5'- ggtctccacaagctcagccttctgagtttggtcctcgc-3' (forward) and 5'-gcgaggaccaaactcagaaggctgagcttgtggagacc-3' (reverse). All primers were purchased from Invitrogen. Retroviruses were generated using BOSC cells and used to infect SK19 cells. Cells were infected with HLA-B-encoding viruses or control viruses lacking HLA-B. Infected cells were selected by treatment with 1 mg/ml G418 (Life Technologies), and maintained in 0.5 mg/ml G418.

## Flow cytometric analysis to assess MHC-I cell surface expression

A total of $1 \times 10^5$–$1 \times 10^6$ cells were washed with FACS buffer (phosphate-buffered saline (PBS), pH 7.4, 1% FBS) and then incubated with W6/32 or HC10 antibodies at 1:250 dilutions for 30–60 min on ice. Following this incubation, the cells were washed three times with FACS buffer and incubated with GαM-PE at 1:250 dilutions for 30–60 min on ice. Following incubations, the cells were washed three times with FACS buffer and analyzed using a BD FACSCanto II cytometer. For peptide occupancy assay, cells were preincubated with peptides for 2 hr at 37°C before staining.

## Cell adhesion assay

SK19 cells expressing HLA-B*35:01, HLA-B*35:01-CD8 null or lacking HLA-B*35:01 (those infected with a control virus lacking HLA-B) were first labeled with CFSE according to manufacturer's protocol. For microscope-based assays, SK19 cells were plated to glass-bottomed petri dish the day before the adhesion assay. After washing the dish with medium, CTLs were added at 1:1 and incubate at 37°C for 2 hr. The cells were then washed with $1 \times$ PBS and fixed with 2% PFA. After staining with anti-CD8-APC, imaged using a Leica SP8 confocal microscope. For flow-cytometry-based assays, CFSE-labeled SK19 cells were incubated in suspension with CTLs at 37°C for 2 hr. The cells were then washed with $1 \times$ PBS, fixed with 2% PFA and stained with anti-CD8-APC. CFSE and CD8 double positive cells were quantified by flow cytometry as conjugated cells.

## HLA-I clustering in the immunological synapse

PBMCs from donor carrying HLA-B*08:01 and B*35:01 were preactivated with PHA to express peptide-deficient conformers of HLA-I. Cognate peptide RL8 (100 uM) or DMSO was loaded at 37°C for 2 hr. The PBMCs and HLA-B*08:01-RL8 specific CTL line (CTL B8-RL8) were mixed, centrifuged briefly and incubated for 10 min at 37°C to allow immunological synapse formation. Cells were fixed and stained with anti-CD8-APC and W6/32-FITC or HC10-FITC antibodies. Cells were imaged using a Leica SP8 confocal microscope. FITC and APC emission were collected in different channels. Data were processed using Leica Imaging software and ImageJ software. The intensity of HLA-I molecules at the interface was compared with the membrane at a noncontact area and plotted as the fold increase above background.

## CD8$^+$ T cell activation assay

Primary CD8$^+$ T cells were purified from PBMCs by negative selection by magnetic-activated cell sorting (MACS, Miltenyi Biotec), according to the instructions. Tetramers were incubated with primary CD8$^+$ T cells or CTL lines at indicated concentrations in serum free medium for 30 min at room temperature and 5 µg/ml anti-CD28 was then added. FBS was supplemented to a final concentration of 10%. $2 \times 10^5$ CD8$^+$ T cells were stimulated at 37°C for six hours in the presence of brefeldin A (Golgiplug, 1:1000; BD Biosciences). Intracellular staining assays were performed to test cytokine expression. Briefly, cells were washed and fixed using 100 µl of 4% formaldehyde at RT for 10 min. After being washed, cells were incubated with 100 µl of 0.2% saponin and then stained with fluorochrome-conjugated antibodies specific for intracellular markers at RT for 30 min. After a final wash, flow cytometry measurements were acquired on a BD FACSCanto II flow cytometer. Flow cytometry gates to identify positive cytokine signal were based on unstimulated control tubes.

## Cell lysis assay

Primary CD4$^+$ T cells were purified from PBMCs by negative selection by magnetic-activated cell sorting (MACS, Miltenyi Biotec), according to the instructions and activated then with PHA. Cells form donors carrying HLA-A*02:01 or HLA-B*08:01 were pulsed for 2 hr with AL9 or RL8 peptides at 100 µM, respectively, together with or without B*35:01 blocking peptides and then incubated with corresponding CTL lines (A2-AL9 or B8-RL8) at indicated ratios for 5 hr at 37°C. Cells were then stained with anti-CD4, anti-CD8 and live/dead marker 7AAD or Aqua to test the viability of CD4$^+$ T cells by flow cytometry.

## Statistical analysis

Statistical analyses (ordinary one-way ANOVA analysis with Fisher's LSD or Dunnett test) were performed using GraphPad Prism version 7.

## Acknowledgements

We thank the NIH tetramer core facility staff for providing the LZ-ELBM HLA-B constructs and Dr. Bruce Walker for the CTL line A2-AL9. We are grateful to all blood donors and the staff at the Michigan Clinical Research Unit (MCRU). We thank Brogan Yarzabek for donor sample collections and preparations for HLA genotyping or immunological assays, Dr. Gayatri Silva for assistance with HLA genotyping of donors, and Eli Olson for generating the CTL line B8-RL8. We thank Dr. Kaushik

Choudhuri and Dr. Fenglei Li for the protocol to detect HLA-I clustering in the immunological synapse and helpful discussions. We thank Dr. Fei Wen for the tetramer staining protocol and helpful discussions. We also thank Dr. Clay Brown, Center for Structural Biology, Life Sciences Institute, University of Michigan for the generation of CD8–encoding plasmid. We thank the University of Michigan DNA Sequencing Core for sequencing analyses, and Elizabeth Smith of the University of Michigan Hybridoma Core for antibody production. This work was supported by the National Institute of Allergy and Infectious Diseases of the National Institutes of Health Grant (R01AI044115 to MR).

## Additional information

### Competing interests
Sujatha Krishnakumar: is affiliated with the Sirona Genomics, where the HLA genotyping for our study was done. The author has no financial interests to declare. The other authors declare that no competing interests exist.

### Funding

| Funder | Grant reference number | Author |
|---|---|---|
| NIH Office of the Director | R01AI044115 | Malini Raghavan |

The funders had no role in study design, data collection and interpretation, or the decision to submit the work for publication.

### Author contributions
Jie Geng, Investigation, Methodology, Writing—original draft, Writing—review and editing; John D Altman, Sujatha Krishnakumar, Methodology, Writing—review and editing; Malini Raghavan, Conceptualization, Supervision, Funding acquisition, Methodology, Project administration, Writing—review and editing

### Author ORCIDs
Jie Geng (iD) http://orcid.org/0000-0001-8722-2228
John D Altman (iD) http://orcid.org/0000-0001-9733-2359
Sujatha Krishnakumar (iD) http://orcid.org/0000-0003-1961-3432
Malini Raghavan (iD) http://orcid.org/0000-0002-1345-9318

### Ethics
Human subjects: Blood was collected from consented healthy donors for HLA genotyping and functional studies in accordance with a University of Michigan IRB approved protocol (HUM00071750).

### Decision letter and Author response
Decision letter https://doi.org/10.7554/eLife.36341.017

## Additional files

### Supplementary files
• Transparent reporting form
DOI: https://doi.org/10.7554/eLife.36341.013

### Data availability
The data that support the findings of this study are openly available in Dryad at https://doi.org/10.5061/dryad.543pp71.

The following dataset was generated:

| Author(s) | Year | Dataset title | Dataset URL | Database, license, and accessibility information |
|---|---|---|---|---|
| Geng J, Altman JD, Krishnakumar S, Raghavan M | 2018 | Empty conformers of HLA-B preferentially bind CD8 and regulate CD8+ T cell function | http://dx.doi.org/10.5061/dryad.543pp71 | Available at Dryad Digital Repository under a CC0 Public Domain Dedication |

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
