## [Decision Letter]

[Editors’ note: minor issues and corrections have not been included, so there is not an accompanying Author response.]

Congratulations, we are pleased to inform you that your article, "Empty" conformers of HLA-B preferentially bind CD8 and regulate CD8^+^ T cell function", has been accepted for publication in *eLife*.

Both reviewers and my reading is that it is a novel study that addresses a long-standing problem. We provisionally accept given that the small correction requested by one reviewer is made. Please make this correction in correspondence with the production staff as the manuscript will require not further editorial oversight.

*Reviewer #1*

MHC class I molecules are key to immune surveillance of virus infected or cancer cells due to their ability to present endogenous peptides on the cell surface. The T-cell receptors of CD8^+^ T cells can then recognize the novel peptide-MHC molecules as flags to initiate appropriate immune responses. Both the TCR and CD8 molecules act in concert to recognize and respond to the peptide loaded MHC molecules. However, under certain circumstances, for example, when the peptide supply is limited by inhibition of TAP, the peptide transporter, MHC class I molecules cannot be loaded with appropriate peptides. Nevertheless, some alleles of the MHC class I molecules (e.g. HLA-B*35:01) can be expressed on the cell surface despite the absence of peptides. What possible function could these empty MHC I molecules serve?

Here, Geng and colleagues describe an unexpected novel function for empty MHC class I molecules. They show that these empty B*35:01 molecules can bind CD8 molecules on a large fraction of all CD8^+^ T cells as well as NK cells. These interactions enhance cell adhesion and clustering within the immunological synapse. If expression of the empty MHC I molecules is reduced by peptide loading it also reduces the ability of CD8^+^ T cells to kill their cognate targets. Thus empty MHC I molecules are functionally capable of providing an additional boost to killer T cell function. The experiments are well designed and well controlled. The conclusions are compelling.

The authors have written an insightful discussion of their unexpected discovery of a function of empty MHC I molecules. For example, many autoimmune and infectious disease outcomes are linked to specific MHC I alleles such as HLA-B*27 in autoimmune ankylosing spondylitis. Because empty HLA-B*27 molecules are expressed on the cell surface, it is possible that these molecules could play a role in this autoimmune disease linked to this MHC I allele.

Overall, this is an excellent manuscript with novel findings and interesting implications.

Minor comment: The legend for Figure 6 has an error in HLA- molecules expressed by the CD4^+^ T cells. HLA-A*35:01 should be corrected.

*Reviewer #2*

In the submitted manuscript the authors establish that stable and empty MHC I conformers of HLA-B35:01, as present on the surface of TAP-deficient transfectant or activated CD4T cells, have a role in CD8^+^ T cells activation. By clustering at the immunological synapse the empty conformers induce a more stable CD8-T cell interaction and favor antigen-specific CTL responses.

The experiments are well performed and controlled. The conclusions supported by the experimental data.

In the big picture, this paper indicates an important role for "empty" MHC I molecule in CD8 immune responses; by engaging CD8 molecules with high affinity the empty conformers further stabilize the MHC-I/peptide TCR interaction. Importantly this binding could promote the stability of low affinity MHC I/peptide ligand and enhance CD8-mediated T cell responses to low affinity MHC I/peptide complexes.